# Validity concerns with the Revised Study Process Questionnaire (R-SPQ-2F) in undergraduate anatomy & physiology students

Staci N. Johnson[1]☯*, Eliza D. Gallagher[2]☯, Anna Marie Vagnozzi[3]

**1** Division of Science, Southern Wesleyan University, Central, SC, United States of America, **2** Engineering & Science Education Department, Clemson University, Clemson, SC, United States of America, **3** School of Mathematical & Statistical Sciences, Clemson University, Clemson, SC, United States of America

☯ These authors contributed equally to this work.
* sjohnson@swu.edu

**Data Availability Statement:** The associated data and R script used in the confirmatory factor analysis can be found at https://github.com/vagnozzia408/r-spq-2f_cfa.

## Abstract

The 20-question Revised Study Process Questionnaire (R-SPQ-2F), which is frequently used to categorize student learning approaches as either *deep* or *surface*, was administered to three sections of Anatomy & Physiology (A&P) courses at a highest research university in the southeastern United States as part of a larger research project. Two hundred thirty-one (231) respondents completed the full survey and 11 participants were recruited to a comparative case study. Initial review of interview transcripts raised concerns about the validity of the R-SPQ-2F results with the population of interest. Interview transcripts were coded using *a priori* codes corresponding to the R-SPQ-2F items, and qualitative and quantitative results were then triangulated. Additional survey responses were collected in a subsequent semester and a confirmatory factor analysis (CFA) was performed using the complete responses from 381 students. The CFA yielded similar or better measures of reliability and fit to the two-factor structure as those in previously reported work by other authors. Nonetheless, findings from triangulation suggest that the R-SPQ-2F was not able to group students by deep and surface approaches to learning in the context of an undergraduate A&P course. In addition, six interviews (3 deep, 3 surface) demonstrated a new theme of *surface leading to deep* with participants indicating that memorization was necessary for the purpose of gaining a full understanding of the course material. This mixed method analysis calls into question whether the results are valid for separating student approaches into the previously published descriptions of deep and surface approaches. The finding of the *surface leading to deep* orientation, which may align with previous descriptions of an *achieving* approach, has significant implications for both research and instruction, as memorizing and other "surface" strategies are often minimized and discouraged, yet are an important step in student learning.

**Funding:** The authors received no specific funding for this work.

**Competing interests:** The authors have declared that no competing interests exist.

# Introduction

Student learning continues to be a topic of interest for educators across many contexts and educational levels. Within this body of literature, student approaches to learning (SAL) research has examined both the affective and contextual aspects of learning to elucidate student cognitive responses to the task of learning [1–3]. The SAL concepts of *deep and surface approaches to learning* [4] have been consistently utilized in educational research over the past 40 years and have more recently been used to understand how the biological subdisciplines of anatomy and physiology are learned, specifically in medical education [5, 6]. Biggs and colleagues first developed the Study Process Questionnaire and Learning Process Questionnaire instruments to describe student approaches to learning in various contexts [2]. The Revised Study Process Questionnaire (R-SPQ-2F), the most recently developed instrument that categorizes student approaches as either *surface* or *deep*, has been used in educational research studies and within physiology education [2, 6, 7].

## Development of SAL theory

Present research on student learning has been built upon findings from the 1970s and 1980s related to student learning approaches and whether these approaches are fixed or context-dependent [8]. The SAL body of literature was established by four main research groups. Because these groups were addressing the same questions during the same period of time, findings from one group influenced the views and responses of the others. The Swedish group, led by Marton, introduced the terms *deep* and *surface approaches* to learning and provided evidence that these approaches were flexible and context-dependent [4]. The findings from the work of the Lancaster and Richmond groups supported and extended the findings of Marton [8, 9]. The Australian group was led by John Biggs and mainly used quantitative methods to understand student approaches to learning. Biggs developed various iterations of the 3P (Presage–Process–Product) learning model, which recognized the inter-relationships of student characteristics, teaching context, student learning processes, and learning outcomes [1, 2]. He also developed multiple iterations of the Study Process Questionnaire (SPQ [10] and R-SPQ-2F [2]) to distinguish between deep and surface learning approaches of students. This instrument categorizes student learning approaches based on their motive for learning and the strategies they utilize. Agreeing with Marton, Biggs held that learning and its approach were context-dependent and flexible [2].

Beattie and colleagues summarize the findings from these groups in this manner:

> Thus this literature, viewed as a whole, demonstrates that a student's approach to learning is only partly a function of his or her general characteristics, since it can be modified by specific learning situations. Such situational influences include the students' perception of the relevance of the learning task, the attitudes and enthusiasm of the lecturer and the expected forms of assessment. The extent to which a student's predilection for a particular approach can be modified is determined by their meta-learning capability.
>
> ([8], pg. 10)

Following in the European and Australian traditions, SAL can be viewed as a process that combines affective traits of the student with the specific learning context. This interaction leads to a specific cognitive response to the task. Overall, the idea of deep and surface approaches to learning was widely adopted in the study of learning in higher education and beyond [6, 7, 11–14]. As research programs moved forward, they began to focus on how to

promote a deep approach to learning, as well as how to assess deep learning approaches in students [8].

## Development of surface and deep approaches to learning

While the terms *deep* and *surface* to describe student learning approaches have been widely used in education research over the past 40 years, the definitions of these terms have been refined but mostly retained from their original introduction. A *deep approach* to learning has been previously defined as "an approach that connects new information to previous relevant knowledge" [8] and is aligned with a focus to gain understanding of meaning and an intention to comprehend [4]. Biggs also connected this approach to the process of *internalizing*, which is defined as an interest in personal growth and an intrinsic motivation to learn.

A *surface approach* to learning has been previously defined as "an approach that focuses on bare essentials and reproduces through rote learning or memorization" [8]. Other characteristics may also include memorization to succeed on a test, retention of literal aspects with no critical analysis or personal contribution, or simply storage of information [4]. Biggs also connected this approach to the process of *utilizing*, which is viewing study as a task to accomplish and overcome in order to pursue a career.

Multiple quantitative measures have been developed that use the terms of *deep* and *surface* to describe student approaches to learning, including the Approaches to Studying Inventory (ASI; [15]), Student Cognitions about Learning (SCALI; [16]) and the Inventory of Learning Styles (ILS; [17]). In a similar effort, Biggs and colleagues developed and then revised the aforementioned Study Process Questionnaire [2]. However, little information was provided about the specific choices for item retention or subscale definitions on the revised instrument. Recent work within psychology has more clearly defined the presence of these types of questionable measurement practices (QMPs) as common within the literature despite the fact that they threaten the validity and conclusions of research [18].

As deep and surface learning approaches were studied in additional cultures and contexts, new questions arose. The simple categorization of a deep or surface approach and the associated motives and strategies failed to capture the approaches taken by all students. A "new" approach to learning that combined understanding and memorization was described and coined as an *achieving* learning approach by Kember [19]. In addition, this work further expanded the 3P Model and focused on how a student's preferred learning approach interacted with the teaching environment to produce learning activities.

Biggs' work in developing his quantitative instruments identified two distinct groupings that interacted with the surface and deep approaches: a student's motive and their strategy [2]. *Motive* is defined as the student's intention toward the work, which may include a fear of failure, intrinsic interest, or achievement. *Strategy* is defined as the particular actions taken by a student and their outcomes, which may include repetition or rote learning. This can also include work to maximize meaning and develop understanding, or an effective use of space and time. These characteristics form the basis for the items on the R-SPQ-2F.

## R-SPQ-2F survey instrument

The R-SPQ-2F instrument provides information about the preferred learning approaches of students [2]. It consists of 20 items that are reported to fall on one of two approach scales or factors (Surface and Deep) and one of two characteristic groups or subscales (Motive and Strategy) [2, 20]. For instance, item 1 (*I find that at times studying gives me a feeling of deep personal satisfaction*) is grouped on the Deep factor and Motive subscale. Overall, five items fall on each of the four factor and subscale combinations. The 20 items are scored using a 5-point

Likert-type scale (A—*this item is never or only rarely true of me* to E—*this item is always or almost always true of me*) and then converted to numerical data (A = 1 to E = 5). Main factors (Surface, Deep) and subscales (Deep Motive, Deep Strategy, Surface Motive, Surface Strategy) are calculated by summing the responses to the corresponding questions. The full survey and complete scoring instructions are available in previous publications [2]. Previous psychometric analysis completed with undergraduate students in the late 1990s found the instrument to have acceptable scale reliability (Cronbach's $\alpha$ = 0.73 for deep and 0.64 for surface) and a reasonable fit to the two-factor structure for the general undergraduate population at that time [2].

Justicia and colleagues [20] examined the underlying structure of the R-SPQ-2F using exploratory and confirmatory factor analysis with survey responses from undergraduate students in Spain. Their results support a two-factor structure as reliable, with the 20 items clustering as noted in the original survey administration instructions. However, their results challenge the ability of the instrument to differentiate subscales. In addition, the validity of the named factors and subscales was not addressed. Entwistle & Entwistle [9] found that a qualitative analysis of student interviews and written responses paralleled a surface and deep approach to learning. However, qualitative data to support the alignment of the two factors measured by the R-SPQ-2F with the SAL constructs of surface and deep learning approaches have not been reported. These gaps in the literature fall within the bounds of QMPs as defined by Flake & Fried [18].

## Reliability and validity in survey research

For survey results to be useful, they must be both reliable and valid. *Reliability* refers to the extent to which results are consistent across different administrations of the instrument and can be evaluated using measures of internal consistency across items, stability over time, and equivalence to other instruments measuring the same constructs [21–23]. There are many components of instrument reliability that might be measured by researchers, but the choice of appropriate measurements depends on the construct under consideration and the specific context. For example, checking for stability over time using a standard test-retest protocol would not be appropriate for an instrument such as the R-SPQ-2F, where individual responses are expected and intended to vary based on context. In this paper, we consider the same aspects of reliability that have been demonstrated by other researchers using the R-SPQ-2F in other settings.

*Validity* is concerned with how well the instrument measures what it is intended to measure. Although reliability is a necessary condition for validity, it is not sufficient to establish the validity of results, which is done through checking content validity, criterion validity, and construct validity [21–27]. *Content validity* refers to the extent to which the instrument covers the entire domain of interest. *Criterion validity* is similar to equivalence, in that it is concerned with the extent to which the items measure the same domain as other instruments with the same intent. Lastly, *construct validity* refers to the extent to which one can draw inferences about the domain based on responses to the instrument. Validity of results from one population does not imply the validity of results from the same instrument used with a different population [22].

## Research question

The R-SPQ-2F was developed and validated with undergraduates from a variety of majors in Hong Kong in the late 1990s. While additional validity and reliability work has been reported in undergraduate populations in various countries and in graduate and professional school

contexts, no work has been published about undergraduate STEM students in the United States. This study addresses the research question: "Does the R-SPQ-2F yield valid results for classifying the learning approaches of STEM undergraduates enrolled in Anatomy & Physiology courses at an R1 institution in the southeastern United States?"

## Methods

This study was conducted as one step in a comparative case study that investigated the cognitive processes and pathways of undergraduate Anatomy & Physiology students. The research was reviewed and approved as exempt by the Institutional Review Board at Clemson University (2018-310).

The quality frameworks of Q3 Quality in Qualitative Research [28] and Legitimation [29, 30] were used to guide the design of this protocol. The Q3 framework provides six areas of validation to consider in all stages of qualitative research, while the Legitimation criteria were used to strengthen the conduct and reporting of mixed methods research [29].

### Quantitative sample & data collection

During a particular fall semester, a total of 824 students were enrolled in three sections of two Anatomy & Physiology courses at a large institution in the southeastern United States classified as "highest research" (R1) by the Carnegie classifications [31]. During the second week of classes, course instructors were emailed text for both an in-class announcement and an email to students. These course instructors were not part of the research team. This invitation included a link to the "Anatomy and Physiology Questions" Survey in Qualtrics [32], comprised of the 20 items that form the R-SPQ-2F followed by prompts for major, current section enrollment, and intent to enroll in the subsequent course in the next semester. The non-R-SPQ-2F items were used as part of the selection process for the full study. Instructors were not provided any information about which students completed the survey or were invited to participate in the full study. Two hundred thirty-one (231) students completed the survey for an overall 27.9% response rate. This low response rate is not unexpected, as survey completion was not a course requirement and few additional incentives were offered.

A pool of potential participants for the full study was created of all respondents who provided informed consent, completed the R-SPQ-2F items, planned to take the second course of the sequence, and self-identified as a STEM or health science major based on two-digit Classification of Instructional Program (CIP) codes [33]. Majors within the bounds of the study were Engineering (code 14), Engineering Technologies and Engineering Related Fields (code 15), Biological/Biomedical Sciences (code 26), Physical Sciences (code 40), or Health Professionals and Related Professions (code 51), although code 15 did not appear in the sample. The remaining pool consisted of 117 students (51.6% of those completing the survey, 14.2% of the course population).

Based on previous literature indicating a lack of inclusion of students with a surface approach preference in education research [15], the intent was to recruit participants who showed a strong preference for either a surface or a deep approach. Because it is possible to receive a high score for both surface and deep learning approaches on the R-SPQ-2F, the difference in Deep and Surface approach scale scores was used as the selection criterion from within the winnowed sample. Fig 1 provides a histogram of deep–surface differential scores ranging from −33 (extreme surface differential) to +29 (extreme deep differential) for the winnowed pool.

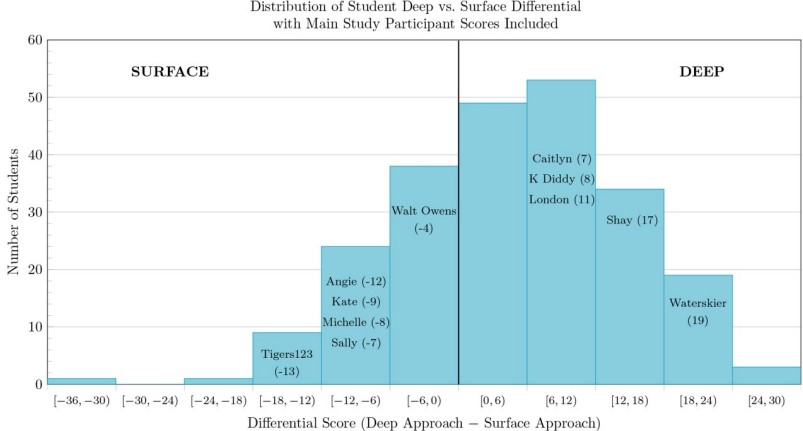

**Fig 1. Distribution of differential scores for students in our sample (*n* = 231), with participant scores highlighted (*n* = 11).**

## Qualitative sample & data collection

The winnowed pool was then divided by course and rank-ordered based on differential scores. Within each class, the participants with the four most extreme differential scores at each end of the scale were invited to the full study for a target sample size of 16 students. If no response was received within two days, a reminder email was sent. After an additional three-day window, the student was removed from the list and the next rank-ordered candidate from that course was invited. The final participant pool for the full study included 11 students, five with a deep approach preference and six with a surface approach preference based on their R-SPQ-2F differential scores. These participants, together with their differential score and self-selected pseudonym, are shown with their relative location in the histogram of differential scores in Fig 1.

Interviews with the 11 participants were scheduled within three weeks of initial completion of the R-SPQ-2F and completed between September 18 and October 3 of the semester in which the study took place. The timing of the interview was important because SAL is considered a flexible characteristic that is impacted by course activities and other items described in the 3P Model [2]. The interview protocol consisted of open-ended questions in a semi-structured protocol to allow participants the freedom to expand or elaborate on their responses. The protocol is provided in Table 1. Prompts were designed to probe for information about

**Table 1. Full semi-structured interview protocol, allowing follow-up questions for clarification of responses to each prompt.**

| | |
|---|---|
| 1. | Describe your A&P class. What do you think about the assignments? Grading procedure? Teaching style? |
| 2. | How is this different from your previous biology physiology courses? |
| 3. | How do you define learning? memorizing? studying? understanding? |
| 4. | How would you rank these (learning, studying, memorizing, understanding) in terms of your personal preference? |
| 5. | How would you rank these (learning, studying, memorizing, understanding) for what is needed for success in your Anatomy & Physiology course? |
| 6. | What do you think is the best approach to learning in this A&P class? (Variable based on response to Q5. Use terms ranked 1 and 2.) |
| 7. | What do you hope to gain from this course? |
| 8. | How do you think the course learning objectives will help you meet your personal goals? (Provide copy of course learning objectives taken from the course syllabus provided by the instructor.) |

teaching context, student characteristics and preferences, and learning process and approach, aligned with the theoretical framework for the full study. It was not a specific intent during this interview to probe for validity of the R-SPQ-2F with this population, so there is not direct alignment between the interview and the survey. Interviews ranged in length from 22 minutes to 33 minutes, with a mean time of 27 minutes.

Process reliability, which provides conditions to make the research process as independent from random influences as possible, was addressed by maintaining the same core prompts for each interview [28]. All interviews were conducted in person and in a neutral location to allow for privacy and quality recording. To support communicative validity and process reliability, interviews were recorded with a digital recorder and transcribed verbatim using Descript software [34] in preparation for analysis. Theoretical validation focuses on the fit between the phenomenon under investigation and the theory produced [28]. In light of this, the interview prompts were designed to expose the reality of the unique learning processes and pathways taken by members within each bounded case. The semi-structured nature of the interview allowed for clarification of student use of words or description of ideas. The Legitimation framework from Onwuegbuzie and colleagues [30] was utilized to ensure quality during the mixing of the data, particularly in the area of weakness minimization.

## Analysis

As interview transcripts were verified, concerns arose in the research team about the ability of the R-SPQ-2F to differentiate surface and deep approaches within this population of undergraduate A&P students. Triangulation of individual item responses with their interview excerpts revealed a lack of agreement between the quantitative and qualitative data. This finding led to detailed analysis comparing quantitative (R-SPQ-2F responses, factor/subscale scores, and differential scores) and qualitative data (interview responses) to answer our research question. Analysis proceeded in four main steps:

1. **Qualitative and quantitative item comparisons:** *A priori* codes for surface, deep, surface to deep, and each of the 20 R-SPQ-2F items were used to identify passages that provided qualitative information relevant to each of the 20 R-SPQ-2F items. *A priori* coding proceeded in iterative stages, with one team member identifying all excerpts that were considered to meet the criteria for a specific *a priori* code and the second team member blind-coding a subset of the data for the same *a priori* code. These iterative cycles continued until the team reached agreement on the boundaries of each code and on the coding of specific passages within the data.

2. **Quantitative and qualitative scale comparisons:** After *a priori* coding was complete, the data were grouped by participant and R-SPQ-2F item. Each member of the research team independently determined whether the available data, considered holistically, indicated agreement or disagreement with the R-SPQ-2F item. Because the R-SPQ-2F is scored on a 5-point, Likert-type scale, a response of 1 or 2 on the survey was considered a "disagreement" with a positively worded item, while a response of 4 or 5 was considered an "agreement." When the quantitative and qualitative data both indicated agreement or both indicated disagreement, we coded this as *alignment*. When one indicated agreement and the other indicated disagreement, we coded this as *misalignment*. When the qualitative data indicated agreement or disagreement and the survey response was a 3 ("neutral"), we coded this as *mild misalignment*.

3. **Item Review:** Each item of the R-SPQ-2F was reviewed by the research team to determine the expected scale (Deep or Surface) and subscale (Motive or Strategy) measured, as well as additional areas of concern for each question.

4. **Confirmatory Factor Analysis:** In addition to the data collected for recruitment to the main study, additional responses to the R-SPQ-2F were collected during the first two weeks of a second fall semester. In total, 381 complete responses and 66 partial responses were collected. An item-level confirmatory factor analysis (CFA) was performed using the complete responses to assess the fit of the previously reported deep and surface approach factor structure [2, 20] to the data from the population of interest and assess the reliability of the instrument.

Each stage of analysis contributed new insights into our concerns with using the R-SPQ-2F with this population of students.

## Research quality considerations

During analysis, qualitative responses were compared to the responses on the quantitative survey. The process of comparing student interview responses to responses to each survey item provided an opportunity for inside-outside legitimation, which is concerned with the extent to which the participant's view is accurately presented and utilized for purposes of explanation and description [30]. The steps for process reliability helped to ensure accurate presentation of participant words. In addition, the research team took care to take participant words at face value when determining alignment between the qualitative and quantitative data. Weaknesses minimization occurred as the qualitative data allowed for a greater breadth of response from participants than the quantitative survey alone [30].

## Results

Findings from the four steps of analysis are presented sequentially.

### Qualitative and quantitative item comparisons

Table 2 provides information about the number of participants who provided a coded excerpt for each R-SPQ-2F item and the total number of excerpts coded for that item. As previously mentioned, the process of comparing qualitative and quantitative data was undertaken in a systematic fashion. An in-depth description of the analysis process for a representative prompt is presented in the following section.

**Example of analysis.**  For item 13 (*I work hard at my studies because I find the material interesting*), nine participants provided information about this survey item with 30 total coded excerpts. This is not surprising, as the intention of the interview was to better understand each student's approach to learning in their Anatomy & Physiology course and this prompt asks for similar information. This item is compound and gives two different statements: 13a) *I work hard at my studies* and 13b) *I find the material interesting*. The coded excerpts were identified by two coding passes completed for this item to capture qualitative information about effort level given by participants in the course (corresponding with statement 13a) and the participant's interest level in the material of the course (corresponding with statement 13b). For compound items such as this one, diagrams like the one shown in Fig 2 were constructed to represent what agreement or disagreement in qualitative terms should translate to on the R-SPQ-2F. Consensus was reached that, if the relevant qualitative excerpts indicated that the participant did believe that they worked hard at their studies and that the participant did find the material interesting, a response to item 13 on the R-SPQ-2F with a "4" or "5" would be

**Table 2. Number of participants and coded qualitative excerpts provided for each SPQ item.**

| SPQ Item | Number of Participants Providing Related Quotes | Number of Excerpts Provided |
|----------|------------------------------------------------|------------------------------|
| 1 | 1 | 1 |
| 2 | 1 | 9 |
| 3 | 10 | 14 |
| 4 | 10 | 26 |
| 5 | 1 | 1 |
| 6 | 4 | 5 |
| 7 | 8 | 38 |
| 8 | 4 | 22 |
| 9 | 3 | 3 |
| 10 | 6 | 11 |
| 11 | 6 | 6 |
| 12 | 8 | 14 |
| 13 | 9 | 30 |
| 14 | 0 | 5 |
| 15 | 5 | 8 |
| 16 | 0 | 0 |
| 17 | 1 | 1 |
| 18 | 7 | 10 |
| 19 | 5 | 7 |
| 20 | 4 | 5 |

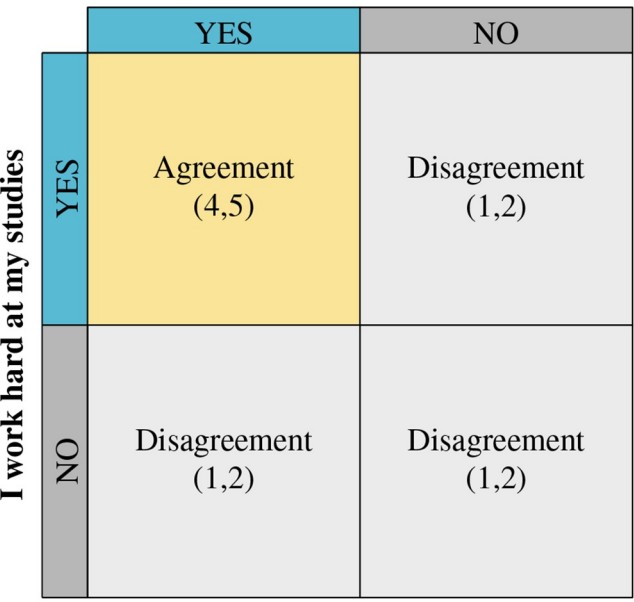

**Fig 2. Diagrammatic representation of agreement or disagreement between interview responses and R-SPQ-2F responses for compound items.**

expected, while any other combination would lead to an expected "1" or "2" in response to item 13.

All coded excerpts for each participant were grouped together and then read as a unit by the research team. The qualitative excerpt(s) were then used to predict an R-SPQ-2F response for each participant. For example, Kate provided the following quotes coded to 13a:

> For Anatomy, **I definitely put a lot more effort into it**. . . And I kind of will compare the two and so I'll look at my big pictures and look at the outline and start looking at those smaller aspects—like maybe the molecules or the compounds and things that are like making up the different materials and all—just try to put things together.

> (emphasis added)

The research team agreed that in these quotes, Kate is expressing that she is working hard in her Anatomy course. For the second portion of this item, Kate provided the following excerpts coded to 13b:

> **I'm really interested in Anatomy** and know it's going to apply to my career. . .[I want to] **understand everything about the human body. I think it's really interesting** and I want to be a physical therapist. So, it's important to know how everything works together and how different people's injuries could affect their anatomy and how that could be treated, so. . .

> (emphasis added)

The research team agreed that these quotes showed that Kate has a strong interest in the course material of her Anatomy course. Because of these quotes, it was predicted that Kate would respond to item 13 on the R-SPQ-2F with a 4 or 5 to signify her agreement with this item. Kate's actual response to item 13 on the R-SPQ-2F instrument was "2". Therefore, the research team classified Kate's qualitative and quantitative responses on item 13 to be *misaligned*.

All 20 items for the R-SPQ-2F were analyzed in the manner described above. Item responses on the R-SPQ-2F that differed by a single unit (e.g. research team prediction = 2, participant response = 3) were considered to be mildly misaligned. Table 3 presents the full results of alignment and misalignment for qualitative and quantitative responses.

In summary, items 3, 6, 9, 10, 13, 17, 18, and 20 were found to present mild concern over misalignment, with stronger concerns regarding items 4, 11, 12, and 19. Items 1, 2, 5, 7, 8, and 15 appear well-aligned. No evidence was available for items 14 and 16. These determinations are based on the number of aligned responses compared to misaligned responses. Items with an equal or greater number of misaligned and mildly misaligned responses present strong concerns. Items with majority alignment but some misalignment are regarded as those with mild concern.

## Quantitative & qualitative scale comparisons

Although 16 of the 20 items had majority alignment, concern remained about the validity of the R-SPQ-2F with this population. In the next stage of analysis, the overall scales of Surface approach and Deep approach were examined. Participant interviews were coded for surface and deep approach themes and these codes were then compared to the Surface and Deep scale scores. The interview transcripts were read again and one of three codes was assigned to

**Table 3. Alignment between qualitative interview data and quantitative SPQ responses for all participants.**

| SPQ Item | Alignment | Mild Misalignment | Misalignment |
|---|---|---|---|
| 1 | 1 | 0 | 0 |
| 2 | 1 | 0 | 0 |
| 3 | 7 | 0 | 3 |
| 4 | 4 | 0 | 6 |
| 5 | 1 | 0 | 0 |
| 6 | 2 | 2 | 0 |
| 7 | 7 | 1 | 0 |
| 8 | 3 | 1 | 0 |
| 9 | 2 | 0 | 1 |
| 10 | 5 | 0 | 1 |
| 11 | 2 | 2 | 2 |
| 12 | 4 | 2 | 2 |
| 13 | 6 | 1 | 2 |
| 14 | 0 | 0 | 0 |
| 15 | 4 | 1 | 0 |
| 16 | 0 | 0 | 0 |
| 17 | 0 | 1 | 0 |
| 18 | 4 | 1 | 2 |
| 19 | 2 | 0 | 3 |
| 20 | 2 | 1 | 1 |

relevant passages as described for the item analysis: *surface*, *surface leading to deep*, and *deep*. Details about the number of excerpts and code definitions are provided in Table 4. While it would be inappropriate to use counts of excerpts to classify the approach of a specific participant or to determine the validity of the R-SPQ-2F, patterns evident in some participant responses may be helpful.

As indicated in Table 4, several participants provided quotes for each of the three codes. Ultimately, most of these groups of quotes have few qualitative differences. For example,

**Table 4. Code definitions, participant differential scores, and number of relevant excerpts coded to each.**

| Participant | Differential | Surface: Quotes indicate a reliance or desire to memorize or rote learn course information | Surface to Deep: Quotes indicate recognition that memorization is necessary with a desire of the participant to understand the material | Deep: Quotes indicate a desire to search for meaning in the task and attain understanding of the material |
|---|---|---|---|---|
| Tigers123 | −13 | 1 | 1 | 1 |
| Angie | −12 | 1 | 0 | 3 |
| Kate | −9 | 1 | 6 | 6 |
| Michelle | −8 | 0 | 0 | 3 |
| Sally | −7 | 2 | 3 | 1 |
| Walt Owens | −4 | 1 | 0 | 4 |
| Caitlyn | 7 | 1 | 0 | 0 |
| K Diddy | 8 | 2 | 1 | 3 |
| Shay | 17 | 3 | 2 | 1 |
| Waterskier | 19 | 1 | 1 | 3 |
| TOTAL | – | 13 | 14 | 25 |

Angie's approach was classified as Surface by the R-SPQ-2F with a differential score of −12. She provided the following quotes which were coded as *surface*:

I think one of the reasons it works out for me this way is because I know that the final exam isn't cumulative. And so that makes me think about the fact that, whenever we end an exam, when we start something new it's going to be the same process. Like I don't have to continue studying what. . .I mean I should, but when it's like new material and I need to just like create more brain space with all these new things. . .

However, she also provided the following quote which was coded as *deep*:

I really hope I learn, and like. . .I guess—is the word sustain? No —with—withhold the information? Right? I don't want to forget it next semester because. . .I'm on a pre-med track. And so I think this is the. . .One of the most more interesting classes I'm going to take—that like, really interests me. Some things that I like, I'm going to see in my future career someday. And so these are concepts that I want to remember and like continue to grow and stuff.

In contrast, K Diddy's approach was classified as Deep by the R-SPQ-2F with a differential of +8. She provided the following quotes which were coded as *surface*:

I feel like right now I'm not like remembering it because it's like "okay, I gotta remember this for the test" and then it's like "okay on to the next thing."

She also provided the following quote which was coded as *deep*:

Like I would prefer to understand it before I start to study the information. . .So I really just wanted to understand. . .Basically how the how the body works. . .And like not a basic understanding because this is not a basic class, but like just enough to help me in my future career.

There is little to no qualitative difference in the description provided by these students in their preferred approach to learning despite a 20-point difference in their deep—surface differential scores. In addition, six participants indicated the need for these approaches to be combined for success in the undergraduate Anatomy & Physiology classroom. The theme of *surface to deep* is demonstrated by the following interview excerpt from Shay, in which she connects the need for memorization in this course context to the understanding of relationships between various parts and systems:

Yeah, for memorizing like you have to know certain terms to be able to build on things. Like if you don't know what like "epithelial" means like—if you don't know that or like the two types of it. . .Then you're not able to apply it. . .So I guess that's uh—like the basis of it. . .And I want to know those terms you're able to know like you're able to like learn them and figure out how they connect together like so. . ."Oh like these two different things are related." So, you know the definition of them and then you know that they were like related then and kind of how they tie together.

Triangulation of qualitative and quantitative results is a standard approach to assessing construct validity of an instrument. The results of this stage of analysis give rise to considerable

concern about construct validity. In light of this, the R-SPQ-2F likely does not measure what it is intended to measure in this population, even if it is successful in doing so with other populations. Overall, this information provides additional evidence that the R-SPQ-2F did not discriminate between the surface and deep learning approaches of students taking an undergraduate Anatomy & Physiology course at the time of this study.

## Item review

The research team reviewed each R-SPQ-2F item to determine our agreement with the factor and subscale to which it was assigned [2]. This review is a standard technique in evaluating face validity, one aspect of content validity for an instrument. Additional areas of concern with those items in the context of interest were also noted. A summary of this analysis is presented in Table 5. Overall, the areas of concern identified with the R-SPQ-2F items can be summarized into four groups:

1. Word interpretation issues

2. Course context/alignment

3. Compound items

4. Validity of factor/subscale description

Word interpretation was an area of concern identified in eight items (1, 2, 3, 4, 8, 9, 10, 11). For several of these, the use of the words "studying," "memorizing," and "understanding" in the prompt was the cause of concern. As noted in the interview protocol in Table 1, students were asked their definition of this term and provided varying responses. These findings are fully discussed in Johnson and Gallagher (in press) or Johnson [35]. Additional terms that may vary in their interpretation due to the nature of the audience include "enough work" (item 2), "pass the course" (item 3), and "learn some things by rote" (item 8). As an example, the term "pass the course" may be defined very differently by students depending on their future goals and aspirations. Consider the following quote from Shay discussing her reasons for taking the course:

> I'm thinking of going to Pharmacy school. And so, this is a prerequisite, like for a lot of Pharmacy schools. Mainly—most of them require both, but some of them just want physiology. But like I mean so I'm gonna be taking both anyway, but it's also on the PCAT too. So like that type of thing, like I need to be prepared for it for that.

For students planning to attend medical school or nursing programs, an A or B in the course may be required when the class is a considered a prerequisite. Therefore, questions remain of how participants may interpret this phrase and it likely varies due to these factors. The phrase "learn some things by rote" is not a common description in the context of this course or population, and this term was never utilized by participants during their interviews. However, it should be noted that the nature of the course content in Anatomy & Physiology requires memorization or rote learning of many terms or anatomical parts for course success.

Four items (4, 5, 6, 9) present concerns related to the specific course context by not being tied specifically to the course in question. For example, item 4 (*I only study seriously what's given out in class or in the course outlines*) is classified as measuring Surface Strategy, but this interpretation would be dependent on the specific expectations for the course in which the survey is completed. For the participants in this study, there is evidence from both the interviews

**Table 5. Results from item review by the research team.**

| SPQ Item | Biggs Factor | Item Area(s) of Concern |
|---|---|---|
| 1. I find that at times studying gives me a feeling of deep personal satisfaction. | Deep Motive | Word Interpretation |
| 2. I find that I have to do enough work on a topic so that I can form my own conclusions before I am satisfied. | Deep Strategy | Word Interpretation, Compound item, Factor/subscale description |
| 3. My aim is to pass the course while doing as little work as possible. | Surface Motive | Word Interpretation |
| 4. I only study seriously what's given out in class or in the course outlines. | Surface Strategy | Word interpretation, Course context/alignment, Factor/subscale description |
| 5. I feel that virtually any topic can be highly interesting once I get into it. | Deep Motive | Course context/alignment, Factor/subscale description |
| 6. I find most new topics interesting and often spend extra time trying to obtain more information about them. | Deep Strategy | Compound item, Course context/alignment, Factor/subscale description |
| 7. I do not find my course very interesting so I keep my work to the minimum. | Surface Motive | Compound item, Factor/subscale description |
| 8. I learn some things by rote, gong over and over them until I know them by heart even if I do not understand them. | Surface Strategy | Word interpretation, Factor/subscale description |
| 9. I find that studying academic topics can at times be as exciting as a good novel or movie. | Deep Motive | Course Context/alignment |
| 10. I test myself on important topics until I understand them completely. | Deep Strategy | Word interpretation |
| 11. I find I can get by in most assessments by memorizing key sections rather than trying to understand them. | Surface Motive | Word interpretation, Factor/subscale description |
| 12. I generally restrict my study to what is specifically set as I think it is unnecessary to do anything extra. | Surface Strategy | Compound item, Factor/subscale description |
| 13. I work hard at my studies because I find the material interesting. | Deep Motive | Compound item, Factor/subscale description |
| 14. I spend a lot of my free time finding out more about interesting topics which have been discussed in different classes. | Deep Strategy | None |
| 15. I find it is not helpful to study topics in depth. It confuses and wastes time, when all you need is a passing acquaintance with topics. | Surface Motive | Compound item, Factor/subscale description |
| 16. I believe that instructors shouldn't expect students to spend significant amounts of time studying material everyone knows won't be examined. | Surface Strategy | Factor/subscale description |
| 17. I come to most classes with questions in mind that I want answering. | Deep Motive | Factor/subscale description |
| 18. I make a point of looking at most of the suggested readings that go with the lectures. | Deep Strategy | None |
| 19. I see no point in learning material which is not likely to be in the examinations. | Surface Motive | None |
| 20. I find the best way to pass examinations is to try to remember answers to likely questions. | Surface Strategy | Factor/subscale description |

and the course syllabi that deep learning or understanding is required for success in the course and on individual assessments. Shay provided this description:

> He gives us the lecture objectives. And he says like if you can fill these out without notes, like and you understand it, like you're able to thoroughly like, write about it, then you'll do well on the tests, I guess.

Therefore, a static assignment of this factor and subscale may not be appropriate and may skew R-SPQ-2F results. Items 5, 6, and 9 are not clearly tied to the course, which seems to violate Biggs' own assertion that student results from the R-SPQ-2F are course- and context-dependent.

Compound items are present for items 2, 6, 7, 12, and 13. In all cases, the items present two statements that are linked, and these statements describe both a strategy and a motive. For example, item 2 can be separated as follows: 2a) *I have to do enough work so that I can form my own conclusions* (strategy) and 2b) *I have to do enough work before I am satisfied* (motive). This pattern is repeated for the other items that are noted and is discussed more fully above in the analysis example of item 13.

The most common area of concern with the R-SPQ-2F items was related to the validity of the factor and subscale descriptions, which was noted in 12 of the 20 items (2, 4, 5, 6, 7, 8, 11, 12, 13, 15, 16, 17, 20). Some of these issues were connected to one or more of the other themes we have previously discussed. When looking at factor or subscale assignment issues, consider the following examples. Item 11 (*I find I can get by in most assessments by memorizing key sections rather than trying to understand them*) is classified as measuring Surface Motive. However, the terms and actions used in this prompt align with a student's strategy toward the course and its material. In addition, items 15 and 16 do not ask for a strategy or a motive, but probe for student or instructor expectation about a course. Item 20 is classified as Surface Strategy. However, the determination of whether this is a deep or surface strategy is dependent on the type of questions utilized by a student, which could be application-based in nature, which would correspond to a deep approach.

## Confirmatory factor analysis

For instruments intended to have multiple factors, such as the R-SPQ-2F, factor analysis provides an accurate measure of instrument reliability by determining the best grouping of items to maximize internal consistency [36]. Because the above analyses presented concerns about the survey validity, and because reliability is a necessary condition for validity, a confirmatory factor analysis (CFA) was performed in R Version 3.5.2 to assess the reliability of the two-factor instrument and its fit to the data gathered from the Anatomy & Physiology students in the study. The MVN and lavaan packages were used for multivariate analysis and confirmatory factor analysis, respectively. This CFA was performed following the procedures outlined in Bandalos [21], and the results reported follow the recommendations of Jackson and colleagues [37]. The objective of the CFA was to determine if the instrument performed at least as reliably for this population of students as it did in previous analyses that form the basis for the justification of its usage in education research [2, 20].

**Data preparation.** A total of 447 responses were obtained. Data collection for Year 1 is described in the Methods section. In Year 2, A&P instructors at six institutions from around the United States sent emails to their classes inviting them to complete the "Anatomy and Physiology Questions" Survey in Qualtrics. The response rate for Year 2 was 28.4% (223 responses from 784 enrolled students). Responses in which students only provided answers for a subset of the items ($n$ = 66) were removed from the dataset via listwise deletion, as the estimation methods in the software packages used for the CFA can only be performed using complete data. This left 381 complete responses to the R-SPQ-2F instrument from Years 1 and 2. For three-factor solutions with three to four variables per factor, Bandalos [21] recommends a sample size of at least 300 if factor loadings are approximately 0.7, and a sample size of 500 or more for lower loadings. The estimates previously obtained through CFA by Justicia and colleagues [20] indicate factor loadings ranging from 0.34 to 0.70, with the majority estimated

between the 0.50–0.65 range. Bandalos further states that more accurate loading estimates are obtained when the number of variables per factor are increased. Because we have ten variables per factor and only two factors, our sample size of 381 was judged to be sufficient for a reasonable model estimation.

Traditional CFA estimation methods were developed under the assumptions of continuous data, univariate and multivariate normality, and the absence of outliers, so these assumptions were tested prior to analysis. Justicia and colleagues critique the fact that prior CFA work on the R-SPQ-2F, including that of Biggs, did not account for the ordinal data generated by the Likert-type instrument items [20]; however, more recent research suggests that as long as there are at least five ordered categories on the item response scales of an instrument, data can be treated as continuous for purposes of model estimation with minimal bias in parameter estimates [21]. Thus, because each item has five levels of response (1–5), the R-SPQ-2F data were treated as continuous.

To assess the data for univariate normality, we considered both the Shapiro-Wilk test and the |2.0| cutoff for item skew and kurtosis recommended by Bandalos. Similarly, for multivariate normality, both Mardia's test and the |3.0| cutoff for Mardia's kurtosis coefficient were considered [21]. Despite that the Shapiro-Wilk test yields strong evidence of non-normality for all twenty items ($p < 0.001$), univariate skew and kurtosis items were all less than |2.0|, indicating that the deviations from univariate normality were not severe. However, Mardia's test indicates deviation from multivarite normality ($p < 0.0001$), and the kurtosis coefficient of 6.87 is well above the |3.0| threshold. As a result, the estimation methods in this CFA were chosen to allow for non-normal data.

Using the MVN package in R, univariate outliers were identified for items 1, 2, 7, 11, 12, 13, 15, 16, and 18, and over forty multivariate outliers were identified for the entire dataset. The CFA model was fit to the data both with and without outliers to determine whether they had a noteworthy effect on the model fit, and while removing outliers resulted in slightly different values for parameter estimates and fit indices, evaluation of the fit indices overall did not change the assessment of the model as a whole. Ultimately, the decision was made to retain the outliers in the dataset, as they represented a reasonable range of student responses from the population of interest. The CFA results presented in this paper reflect those for the full dataset with no outliers removed.

The covariance matrix of item responses was utilized as the input matrix for the CFA. The corresponding Pearson correlation matrix and standard deviations are provided in Table 6 for those wishing to replicate our analysis. Note that, with the exception of the correlation between items 10 and 14, all correlations between items on the same scale (Deep or Surface) are at least 0.1. Correlations between items on different scales are all less than 0.1 and, in many cases, negative. In general, we see a relationship between the items we would expect to be correlated based on the nature of the instrument.

**Model specification.** The model tested in this CFA is that reported by Justicia and colleagues [20], which consists of the two factors hypothesized to represent Deep and Surface approaches to learning, but not the Motive and Strategy subscales. In this model, ten of the twenty Likert-type items are hypothesized to load onto the Deep approach factor, while the remaining ten correspond to the Surface approach factor. A description of which items are associated with which factors is included in Table 5, and the graphical representation of the model is shown in Fig 3. Alternative models were not tested; while conducting further analyses to explore the existence of better-fitting models would likely be beneficial, doing so was outside the scope of this study. The purpose of this CFA was solely to assess the fit and reliability of the *existing* instrument structure for the population of interest as compared with the results provided by Biggs [2] and Justicia and colleagues [20].

**Table 6. R-SPQ-2F item correlations and standard deviations.** Hypothesized item correlations are highlighted.

| Item | SPQ1 | SPQ2 | SPQ3 | SPQ4 | SPQ5 | SPQ6 | SPQ7 | SPQ8 | SPQ9 | SPQ10 | SPQ11 | SPQ12 | SPQ13 | SPQ14 | SPQ15 | SPQ16 | SPQ17 | SPQ18 | SPQ19 | SPQ20 |
|---|---|---|---|---|---|---|---|---|---|---|---|---|---|---|---|---|---|---|---|---|
| SPQ1 | 1.000 | | | | | | | | | | | | | | | | | | | |
| SPQ2 | 0.458 | 1.000 | | | | | | | | | | | | | | | | | | |
| SPQ3 | −0.107 | −0.083 | 1.000 | | | | | | | | | | | | | | | | | |
| SPQ4 | −0.007 | −0.032 | 0.345 | 1.000 | | | | | | | | | | | | | | | | |
| SPQ5 | 0.209 | 0.209 | −0.056 | −0.008 | 1.000 | | | | | | | | | | | | | | | |
| SPQ6 | 0.287 | 0.285 | −0.042 | −0.012 | 0.400 | 1.000 | | | | | | | | | | | | | | |
| SPQ7 | −0.216 | −0.172 | 0.378 | 0.175 | −0.150 | −0.072 | 1.000 | | | | | | | | | | | | | |
| SPQ8 | 0.051 | 0.001 | 0.123 | 0.172 | 0.003 | 0.054 | 0.102 | 1.000 | | | | | | | | | | | | |
| SPQ9 | 0.319 | 0.221 | −0.084 | −0.101 | 0.400 | 0.470 | −0.085 | 0.043 | 1.000 | | | | | | | | | | | |
| SPQ10 | 0.355 | 0.343 | −0.146 | −0.079 | 0.154 | 0.238 | −0.198 | 0.123 | 0.233 | 1.000 | | | | | | | | | | |
| SPQ11 | −0.093 | −0.087 | 0.382 | 0.230 | 0.000 | −0.040 | 0.228 | 0.358 | −0.070 | −0.072 | 1.000 | | | | | | | | | |
| SPQ12 | −0.104 | −0.183 | 0.339 | 0.339 | −0.032 | −0.091 | 0.309 | 0.215 | −0.119 | −0.134 | 0.482 | 1.000 | | | | | | | | |
| SPQ13 | 0.323 | 0.206 | −0.182 | −0.137 | 0.319 | 0.381 | −0.160 | 0.011 | 0.423 | 0.318 | −0.146 | −0.124 | 1.000 | | | | | | | |
| SPQ14 | 0.208 | 0.171 | −0.023 | −0.124 | 0.303 | 0.475 | 0.019 | 0.033 | 0.455 | 0.077 | −0.042 | −0.079 | 0.360 | 1.000 | | | | | | |
| SPQ15 | −0.191 | −0.133 | 0.370 | 0.230 | −0.005 | −0.007 | 0.413 | 0.173 | −0.071 | −0.309 | 0.322 | 0.339 | −0.131 | 0.129 | 1.000 | | | | | |
| SPQ16 | −0.038 | −0.077 | 0.267 | 0.219 | −0.107 | −0.160 | 0.129 | 0.134 | −0.150 | −0.059 | 0.199 | 0.193 | −0.181 | −0.138 | 0.198 | 1.000 | | | | |
| SPQ17 | 0.122 | 0.184 | −0.091 | −0.067 | 0.215 | 0.291 | −0.018 | 0.064 | 0.200 | 0.209 | 0.009 | −0.069 | 0.214 | 0.289 | 0.051 | −0.024 | 1.000 | | | |
| SPQ18 | 0.249 | 0.229 | −0.195 | −0.126 | 0.164 | 0.268 | −0.165 | 0.080 | 0.246 | 0.332 | −0.110 | −0.171 | 0.288 | 0.217 | −0.125 | −0.001 | 0.404 | 1.000 | | |
| SPQ19 | −0.055 | −0.095 | 0.261 | 0.214 | −0.171 | −0.168 | 0.273 | 0.188 | −0.169 | −0.064 | 0.255 | 0.338 | −0.197 | −0.123 | 0.205 | 0.477 | −0.022 | −0.040 | 1.000 | |
| SPQ20 | −0.097 | 0.017 | 0.257 | 0.279 | 0.027 | −0.012 | 0.275 | 0.281 | −0.070 | −0.075 | 0.415 | 0.332 | −0.079 | −0.026 | 0.253 | 0.262 | 0.029 | −0.037 | 0.278 | 1.000 |
| SD | 1.078 | 1.054 | 1.167 | 1.153 | 1.211 | 1.056 | 0.965 | 1.117 | 1.165 | 0.992 | 1.061 | 1.061 | 0.887 | 1.112 | 0.983 | 1.188 | 1.015 | 1.147 | 1.137 | 1.189 |

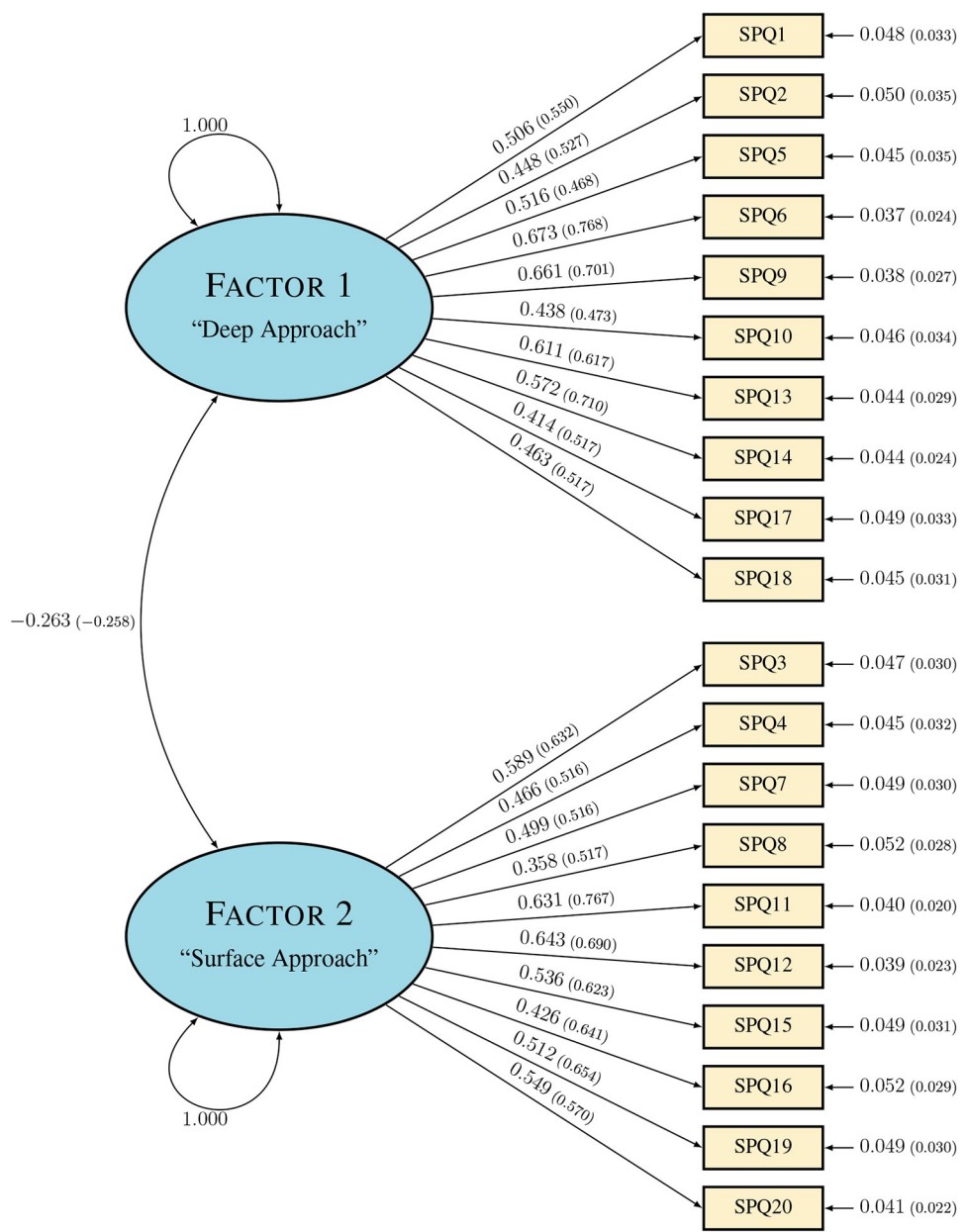

**Fig 3. The hypothesized two-factor structure of the Revised Study Process Questionnaire.**

**Model identification.**   This model meets the requirements for identification as described by Bollen [38], as it has the following characteristics: (1) ten items load onto each of the two factors, which is greater than the minimum requirement of three; (2) each item loads onto only one factor (either Deep *or* Surface); and (3) we assume the measurement error variances to be uncorrelated. Further, we set the factor metric by fixing the mean and variance of the factor "scores" to zero and one, respectively, which allows us to interpret the completely standardized factor loading estimates as the number of standard deviations that an item score would change for a one standard deviation change in the factor. These specifications result in an over-identified model with 169 degrees of freedom.

**Estimation of model parameters.** Two natural choices for model estimation arise, given the characteristics of the data. The first is *weighted least squares* (WLS), which Justicia and colleagues [20] use in the study that led to the two-factor structure of the instrument used predominantly in education research. WLS estimation is advantageous because it makes minimal assumptions about the distribution of the observed variables, and thus the violation of multivariate normality for the R-SPQ-2F data does not pose an issue [38]. In fact, Justicia and colleagues critique Biggs' and other researchers' appearance to ignore the non-normality of the data in prior factor analyses conducted for the R-SPQ-2F [20]. However, in order to be most informative, WLS requires large sample sizes upwards of 2,000 sample points; research shows that, if the sample size is too small, WLS estimation can result in biased parameter estimates, inaccurate standard errors, and a poor fit to the data [21]. It is of note that Justicia and colleagues [20] do not consider this limitation of WLS estimation in their study ($n$ = 522).

An alternative approach, recommended by Bandalos for when data are non-normal and large sample sizes are not available [21], is to use the more traditional *maximum likelihood* (ML) estimation and apply Satorra-Bentler (S-B) adjustments, which correct for the tendency of non-normality to inflate the chi-square goodness-of-fit statistic and underestimate parameter standard errors [37].

In our CFA, we assessed the model fit by considering results from both the WLS and ML estimation methods. The maximum likelihood approach is preferred, based on the recommendations of Bandalos. However, the WLS approach was conducted alongside it to compare with the results obtained from Justicia and colleagues [20]. Completely standardized parameter estimates for factor loadings and standard errors using both approaches are displayed in Fig 3, with the preferred ML estimates in large text and the comparative WLS estimates in parentheses and in smaller text beside them. We see that factor loadings are higher and standard errors are lower when using WLS estimation; however, this should be considered cautiously in light of the small sample size. Still, the ML estimates indicate loadings of 0.35 or higher for all R-SPQ-2F items, similar to those reported by Justicia and colleagues [20].

Also of interest are the $R^2$ values for each item. For the completely standardized estimates, these values can be computed by squaring the estimated loading of each item and are shown in Table 7. Each of these values can be interpreted as the proportion of variance in the item response that can be accounted for by the factor. The ML approach estimates that the first factor, hypothesized to be the Deep approach, accounts for 17.1% to 45.3% of the variance in item responses, and the second factor, or hypothesized Surface approach factor, accounts for 12.8% to 41.3% of the variance in item responses. Weighted least squares $R^2$ estimates are also included for comparison.

**Model testing.** As is fairly common in CFA research despite controversy over its usefulness, chi-square goodness-of-fit tests were conducted for each estimation method. In this test, the null hypothesis is that the model is a good fit to the data, so we hope to see *p*-values greater than a significance level of 0.05 when assessing the fit of a hypothesized model. However, given the dependency of the chi-square test on sample size and its tendency to reject the null hypothesis even when a model fits well (i.e., an inflated probability of a Type I error), Bandalos advocates for assessing a model using multiple fit indices to account for the chi-square test's shortcomings. Similarly, Jackson and colleagues strongly recommend the inclusion of several fit indices and for the cutoff values for each fit index to be specified *a priori* [37].

For this analysis, cutoffs based on prior research were chosen for the comparative fit index (CFI), Tuck Lewis index (TLI), root mean square error of approximation (RMSEA), and standardized root mean square residual (SRMR). Hu and Bentler suggest that CFI and TLI values of 0.95 or higher indicate good fit of a model, while values between 0.90 and 0.95 indicate

**Table 7. $R^2$ values for each of the twenty items on the R-SPQ-2F.**

| Item | $R^2$ Estimates (ML) | | $R^2$ Estimates (WLS) | |
|---|---|---|---|---|
| | Factor 1 (Deep) | Factor 2 (Surface) | Factor 1 (Deep) | Factor 2 (Surface) |
| SPQ1 | 0.256 | | 0.303 | |
| SPQ2 | 0.201 | | 0.278 | |
| SPQ5 | 0.266 | | 0.219 | |
| SPQ6 | 0.453 | | 0.590 | |
| SPQ9 | 0.437 | | 0.491 | |
| SPQ10 | 0.192 | | 0.224 | |
| SPQ13 | 0.373 | | 0.381 | |
| SPQ14 | 0.327 | | 0.504 | |
| SPQ17 | 0.171 | | 0.267 | |
| SPQ18 | 0.214 | | 0.267 | |
| SPQ3 | | 0.347 | | 0.399 |
| SPQ4 | | 0.217 | | 0.266 |
| SPQ7 | | 0.249 | | 0.266 |
| SPQ8 | | 0.128 | | 0.267 |
| SPQ11 | | 0.398 | | 0.588 |
| SPQ12 | | 0.413 | | 0.476 |
| SPQ15 | | 0.287 | | 0.388 |
| SPQ16 | | 0.181 | | 0.411 |
| SPQ19 | | 0.262 | | 0.428 |
| SPQ20 | | 0.301 | | 0.325 |

acceptable fit [21, 39]. For RMSEA and SRMR, values ≤0.05 are indicative of good model fit, while values ≤0.08 indicate moderate but acceptable fit [21, 39, 40].

Fit indices were generated for the R-SPQ-2F model under each type of estimation and are displayed in Table 8.

Internal consistency of the items as they relate to the Deep and Surface approach factors was assessed by computing McDonald's omega for each factor using the `semTools` package in `R`. Coefficients omega are reported in Table 9 for each estimation method.

## Discussion

These results yield some concerns over the reliability of the R-SPQ-2F when the instrument is administered to undergraduate A&P students. However, as discussed below, the results are *at*

**Table 8. Fit indices for the R-SPQ-2F model.**

| Fit Index | ML* | WLS |
|---|---|---|
| Chi-Square | 471.643 | 634.366 |
| Degrees of Freedom | 169 | 169 |
| P-Value | 0.000 | 0.000 |
| CFI | 0.801 | 0.558 |
| TLI | 0.777 | 0.503 |
| RMSEA | 0.069 | 0.085 |
| SRMR | 0.072 | 0.102 |

*All fit indices calculated using S-B adjustments.

**Table 9. Coefficient omega reliability estimates.**

| Factor | ML | WLS |
|---|---|---|
| Factor 1 (Deep) | 0.798 | 0.840 |
| Factor 2 (Surface) | 0.788 | 0.857 |

*least as reliable* with this population as with other populations of interest where reliability estimates have been reported. There are much stronger concerns over the validity of the results of the R-SPQ-2F when administered to undergraduate A&P students.

At best, our quantitative analysis yields moderate reliability of results with this population. As indicated in Table 8, the chi-square test results under each type of model estimation indicate strong evidence ($p < 0.001$) that the model is not a good fit for the data. Looking to the alternative fit indices, we see that none of those calculated for the model using the WLS estimation method indicate a good fit. The "best" results are shown for the ML estimation method with the S-B adjustments for non-normality. Though the CFI and TLI fits do not indicate a good model fit, RMSEA and SRMR both indicate fit index values that correspond to "acceptable" model fits with respect to the index cutoffs specified *a priori*. For comparison, we consider the results of the confirmatory factor analysis by Justicia and colleagues; when taking the preferred maximum likelihood approach, our CFI values are worse than the those found by the authors in their assessment of the R-SPQ-2F (0.92 for their preferred model), but our RMSEA and SRMR indices are slightly better (Justicia and colleagues reported RMSEA = 0.07 and SRMR = 0.09) [20]. Though this is certainly not evidence of a "good" model fit, the model can be deemed *at least* as acceptable as that reported by Justicia and colleagues.

Though the analysis of the R-SPQ-2F instrument by Justicia et al. does not report measures of internal consistency, Biggs reports Cronbach's alpha scores of 0.73 for the Deep approach factor and 0.64 for the Surface approach factor [2]. McDonald's omega is argued to be a similarly interpreted but more accurate measure of internal consistency than Cronbach's alpha when performing confirmatory factor analysis on an instrument such as the R-SPQ-2F [21]. In this regard, using either estimation method, our CFA indicates *more* internal consistency and thus better reliability with our population of interest than the results presented by Biggs, whose Cronbach alpha scores were deemed acceptable.

Taken holistically, the confirmatory factor analysis indicates that the R-SPQ-2F instrument is *at least as reliable* with this population of undergraduate A&P students as it is reported to be in the studies by Biggs and Justicia and colleagues with their populations of interest. Because reliability is a necessary condition for validity, should the reliability of the survey be deemed insufficient by some standards based on the CFA results, the validity of the instrument would justifiably be called into question. However, the R-SPQ-2F has been used and continues to be used in education research. This continued use indicates acceptance, either explicitly or implicitly, of the reliability of the instrument based on the results reported by Biggs [2] and Justicia et al. [20].

Even if one accepts the results of the survey as reliable based on this standard, we nonetheless have reason to believe that the two factors measured by the instrument do not truly represent deep and surface approaches to learning, calling into question the validity of the instrument, at least with this population and potentially with other populations as well. This is cause for concern given the continued usage of the R-SPQ-2F in education research [7, 11, 12, 41, 42], and in A&P education research in particular [43].

Results from the qualitative and quantitative item comparisons yielded eight items with mild misalignment concerns and four items with significant concerns (12/20 or 60% of

items). The comparison of qualitative and quantitative scales raised concern as evidence was present that student learning approaches were not distinguished by the R-SPQ-2F instrument. The review of all 20 items produced concerns in all but three items, and several of the items with concerns had multiple issues. The research team identified word interpretation issues, interactions between the course context and phrasing of items, presence of compound items, and items assigned to a specific factor or subscale that call into question construct validity.

One possible explanation for the issues observed with the R-SPQ-2F in this study is the lack of recognition of the *achieving* approach to learning which has been previously noted in the literature. Kember defined an achieving approach as "an approach that believes memorization is necessary to maintain a high grade, but desires to connect new information to previous knowledge" [19]. As previously mentioned, many of the participants of this study expressed aspirations to attend professional or graduate school. This fact motivated them to achieve high grades while they desired to make additional connections to their existing knowledge. Biggs and colleagues briefly acknowledge this orientation in relation to the original SPQ, stating that "higher order factor analyses [of the original SPQ] usually associate the achieving motive and strategy with the deep approach" [2]. However, the data presented in this paper would question whether this association is true for the updated instrument and for this population. In fact, participants who qualitatively described a learning approach in alignment with the achieving definition were **not** consistently categorized by the R-SPQ-2F as adopting a deep approach to learning.

Another factor to consider related to the validity of this instrument with undergraduate A&P students is the nature of the discipline itself. The participants in this study noted multiple times the need to memorize certain aspects of the course material (classified as a surface approach within the SAL literature) in order to be able to fully understand it. We categorized these responses as *surface to deep* approaches within the qualitative data. Michael and colleagues [44] note that physiology is difficult for students to learn, partly because of the need for an adequate knowledge base or other prerequisite knowledge. Much of this knowledge, like names and locations of anatomical parts or various terms, can only be learned through processes or strategies that are often categorized by instructors and researchers as surface approaches. Given this information, it may also be helpful to consider the surface, achieving, and deep approaches to learning not only as context-dependent characteristics, but perhaps as traits on a continuum rather than as discrete categories or groupings.

Finally, the possible issues mentioned above may stem from employment of QMPs mainly relating to construct validity of the factors and subscales in the development of the R-SPQ-2F. Biggs and colleagues [2] provide no information concerning the revision or retention of items from the longer Study Process Questionnaire. As stated by Flake and Fried:

> As such, modifications. . .introduce uncertainty about the construct validity evidence for the interpretation of the scale score.

([18], pg.16)

As mentioned previously, there is no published information providing qualitative data to support the construct definitions as connected to the items present on the R-SPQ-2F. This gap in the literature makes it difficult to determine the overall validity of this instrument in categorizing student approaches to learning for any population. Our work indicates that the instrument does not produce valid results for the specific population of undergraduate A&P students.

## Conclusion

Although the R-SPQ-2F is widely used and results are accepted as reliable based on previously published measures of internal consistency, questions about the validity of the instrument remain. Validity issues could be attributed to several causes, including omission of the achieving approach to learning, specific features of the biological subdisciplines of anatomy and physiology, or the employment of QMPs in the development of the instrument. Further study is needed to determine the identity of specific factors measured by the R-SPQ-2F.

## Limitations

This work did not begin with the intent to analyze the validity and reliability of the R-SPQ-2F. The interview protocol did not probe directly for answers to the survey prompts, so important ideas and themes from the instrument may not have been detected. However, care was taken to interpret participant words at face value and only declare a misalignment when the qualitative data presented a clear disagreement with the survey prompt.

## Future work

As previously mentioned, researchers or practitioners who wish to utilize the R-SPQ-2F should test the validity of the instrument in their population of interest prior to use. These analyses should include testing for face validity and construct validity. Alternatively, an updated instrument that measures or categorizes student learning approaches as surface or deep could be developed for populations for which the R-SPQ-2F is not valid. Given the minimally acceptable reliability of the two-factor structure of the R-SPQ-2F, additional study could clarify the constructs that are being measured. In addition, it would be worthwhile to evaluate other theoretical factor structures to test alternative models through exploratory and confirmatory factor analyses, considering the overall poor fit to the data in this study. Finally, future work could evaluate the impact of student approach to learning in specific course contexts with revised or redeveloped instruments. This could involve a study of specific teaching tools or practices in undergraduate courses.

## Supporting information

**S1 Data.**
(TXT)

## Acknowledgments

The authors wish to thank the 11 participants of the mixed methods study, students who completed the "Anatomy & Physiology Questions" survey, and instructors who facilitated communication with potential participants. Some results from this manuscript were previously presented. Results from the quantitative and qualitative scale comparison were presented as an oral presentation at the Gordon Research Seminar on Undergraduate Biology Education Research in June 2019. Findings for R-SPQ-2F item 13 were presented as a poster at the Gordon Research Conference on Undergraduate Biology Education Research in June 2019. Findings for R-SPQ-2F item 4 were presented as a poster at the Society for the Advancement of Biology Education Research (SABER) Annual Meeting in July 2019. Preliminary findings from the CFA were presented as a poster at the Tennessee STEM Education Research Conference in January 2020.

## Author Contributions

**Conceptualization:** Staci N. Johnson, Eliza D. Gallagher.

**Data curation:** Staci N. Johnson, Eliza D. Gallagher.

**Formal analysis:** Staci N. Johnson, Eliza D. Gallagher.

**Investigation:** Staci N. Johnson, Eliza D. Gallagher, Anna Marie Vagnozzi.

**Methodology:** Staci N. Johnson, Eliza D. Gallagher, Anna Marie Vagnozzi.

**Project administration:** Staci N. Johnson.

**Resources:** Staci N. Johnson.

**Software:** Staci N. Johnson.

**Supervision:** Eliza D. Gallagher.

**Validation:** Staci N. Johnson, Anna Marie Vagnozzi.

**Visualization:** Staci N. Johnson, Anna Marie Vagnozzi.

**Writing – original draft:** Staci N. Johnson, Anna Marie Vagnozzi.

**Writing – review & editing:** Staci N. Johnson, Eliza D. Gallagher, Anna Marie Vagnozzi.

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
