## [Decision Letter · Decision Letter 0]

2 Oct 2020

PONE-D-20-22094

Validity concerns with the Revised Study Process Questionnaire (R-SPQ-2F) in undergraduate anatomy and physiology students

PLOS ONE

Dear Dr. Johnson,

Thank you for submitting your manuscript to PLOS ONE. After careful consideration, we feel that it has merit but does not fully meet PLOS ONE’s publication criteria as it currently stands. Therefore, we invite you to submit a revised version of the manuscript that addresses the points raised during the review process.

We look forward to receiving your revised manuscript.

Kind regards,

Francesca Chiesi

Academic Editor

PLOS ONE

Reviewers' comments:

Reviewer's Responses to Questions

**Comments to the Author**

1. Is the manuscript technically sound, and do the data support the conclusions?

Reviewer #1: Yes

2. Has the statistical analysis been performed appropriately and rigorously? 

Reviewer #1: Yes

3. Have the authors made all data underlying the findings in their manuscript fully available?

Reviewer #1: Yes

4. Is the manuscript presented in an intelligible fashion and written in standard English?

Reviewer #1: Yes

5. Review Comments to the Author

Reviewer #1: First of all I would like to congratulate the authors for the article, well structured and articulated. The paper is very exhaustive and complete. However, despite its good quality, I would like to suggest some minor issues, in my opinion, easy to adjust.

1) The article is very long. It is clear that as it contains both qualitative and quantitative analyzes, the results section requires a lot of space; however, in my opinion, there is a lot of methodological and procedural information in the result, to make the reading complex and difficult to follow. I suggest the authors to shorten the following sections (and evaluate to integrate discussion):

- Development of SAL theory [shorten lines 16-44; evaluate to delete lines 45-53]

- Development of Surface and Deep Approaches to Learning [lines 76-88]

- Analysis [delete lines 237-244; shorten lines 254-271]

- Examples of Analysis [the section is very long. Evaluate to shorten in not more than 20-25 lines]

- Model specification [lines 563-573 should be maybe moved to data analysis]

2) I suggest the author to spend some lines in a small review of specific recent studies regarding medical students. I feel that the authors should add 20 new recent literature contributes regarding medical students at least and new theoretical perspectives (e.g. Variation theory and others) regarding student approaches.

3) In my opinion one aspect lacking is the contents concerning the professional teaching practice and the implications and applications of the instrument. Discussion is short and not did fully discuss results.

4) The research question s too limited. Please, try to better develop a rationale and a literature review that better explain the research gap that the study wants to fill in and the research questions.

5) Try to explain why the response rate is so low.

6) Since the article is very long, I try to suggest to delete some figure (figure 1, 2 and 3).

Thank you for this opportunity.

6. PLOS authors have the option to publish the peer review history of their article (what does this mean?). If published, this will include your full peer review and any attached files.

Reviewer #1: **Yes: **Massimiliano Barattucci

---

## [Author Response · Author response to Decision Letter 0]

21 Dec 2020

We have uploaded the requested documents for resubmission of our article titled "Validity concerns with the Revised Study Process Questionnaire (R-SPQ-2F) in Undergraduate Anatomy & Physiology Students" (PONE-D-20-22094). Our research team has carefully reviewed the comments and suggestions. We appreciate the time and insights provided by these comments and believe the manuscript is now stronger with incorporation of many of these suggestions. Our 'Response to Reviewer' letter provides a table with detailed information about how we have responded to each point. We appreciate this feedback to make our manuscript stronger.

---

## [Editor Report · Decision Letter 1]

6 Jan 2021

PONE-D-20-22094R1

Validity concerns with the Revised Study Process Questionnaire (R-SPQ-2F) in undergraduate anatomy and physiology students

PLOS ONE

Dear Dr. Johnson,

Thank you for submitting your manuscript to PLOS ONE. After careful consideration, we feel that it has merit but does not fully meet PLOS ONE’s publication criteria as it currently stands.

I accurately read the revised manuscript and checked the changes made in response to the reviewer’s comments. Whereas I can appreciate the effort made to improve the paper, I have two main concerns about some psychometric issues.

1. The CFA fit indices are clearly very poor and not moderate. Did you test any different model? I suggest looking at the modification indices or to exclude some items to identify a better factor solution.

2. I’m not sure I can understand how the factors can be deemed reliable but not valid:

p. 32, line 755 - “We thus have minimal evidence that the two-factor structure of the R-SPQ-2F is a moderately reasonable fit to the data for this population, which supports the R-SPQ-2F reliably measuring two distinct factors. This would agree with the findings of Stes and colleagues [*By the way, Stes et al.’s findings are not described in the introduction because have been deleted*]. However, in light of the results from the other components of this study, we have reason to believe that the two factors measured by the instrument may not truly represent deep and surface approaches to learning, calling into question the validity of the instrument.

p. 34, line 815 - “While the reliability of the two-factor structure of the R-SPQ-2F has been deemed acceptable, questions about the validity of this instrument remain”.

Usually reliability is measured using Cronbach’s alpha or McDonald’s Omega, but this is not the case. Hence, I suggest to specify how reliability was assessed.

Therefore, we invite you to submit a revised version of the manuscript that addresses the points raised above.

We look forward to receiving your revised manuscript.

Kind regards,

Francesca Chiesi

Academic Editor

PLOS ONE

---

## [Author Response · Author response to Decision Letter 1]

8 Feb 2021

Response to Reviewer Comment 1: Our characterization of CFA fit is based on published and accepted fit index cut-offs (Bandalos 2018; Browne 1993; Hu 1999 - see letter for full citations). In addition, while some fit indices are worse than those reported by Justicia and colleagues, two indices (RMSEA and SRMR) were a slight improvement over those reported in their analysis and did provide some minimal evidence of reliability. We have included the citations in the manuscript (p. 26-27) and clarified language within the Abstract (p. 1-2) and Discussion section (p. 27-28) to indicate that the model fits moderately at best. 

 We performed a preliminary exploratory factor analysis, and the model yielded by the 2-factor approach had acceptable factor loadings with the 20 items grouping as reported by Biggs and Justicia. This is not reported, as it is beyond the scope of this study and would increase the length of this paper. We agree that additional models should be tested and have updated the Future Work section (p. 30) to indicate this. 

In Response to Reviewer Comment 2: We thank you for catching that our reference to Stes and colleagues no longer makes sense following our edits of the Introduction. We have addressed this issue. To add clarity to the discussion of reliability and validity, we have added a section in the Introduction (p.6) regarding validity and reliability of survey instruments. This section briefly describes important aspects of reliability (stability, internal consistency, equivalence) and validity (content validity, construct validity, criterion validity) and distinguishes them. We have then made more explicit connections between the evidence and specific aspects of reliability and validity at appropriate points in the Analysis (p.17: Quantitative and qualitative scale comparison; Item review) and Discussion (p.28,29) sections. We also particularly address why we chose to carry out factor analysis rather than using Cronbach's alpha or Kuder-Richardson 20 to assess internal consistency of the R-SPQ-2F (p.21). We have added appropriate citations to support these additional details.

---

## [Editor Report · Decision Letter 2]

17 Feb 2021

PONE-D-20-22094R2

Validity concerns with the Revised Study Process Questionnaire (R-SPQ-2F) in undergraduate anatomy and physiology students

PLOS ONE

Dear Dr. Johnson,

Thank you for submitting your manuscript to PLOS ONE. After careful consideration, we feel that it has merit but does not fully meet PLOS ONE’s publication criteria as it currently stands. Therefore, we invite you to submit a revised version of the manuscript that addresses the points detailed below.

I accurately read the revised manuscript and checked the changes made in response to my comments. Whereas I can again appreciate the effort made in revising the paper, unfortunately I’m not totally convinced that the revisions help to improve the paper or to solve my previous concerns.

Overall, there is still a mixed use of two distinct concepts: reliability and validity. To the best of my knowledge, reliability refers to “internal consistency” and validity refers to the “internal structure” of a scale. If the internal structure (or factor structure) is not supported by the CFA, to discuss reliability is useless.

In details:

1. “Our characterization of CFA fit is based on published and accepted fit index cut-offs [1, 2, 3].” The fact that similar indices have been published do not mean that they are acceptable (criteria are clearly reported by the Authors at p. 27, line 611-612).

2. ML and WLS estimation methods: fit indices are not acceptable in both cases, even if slightly better for ML. However, 11 out of 20 factor loadings are <.30 if ML estimation is used. This result confirms the inadequacy of the tested factor solution.

3. At p. 6, the Section Reliability and Validity is confounding: a) “Reliability is evaluated using measures of internal consistency”: why these measures were not calculated? b) reliability has nothing to do with the “equivalence to other instruments measuring the same constructs”. This aspect concerns validity. c) Criterion validity and construct validity definitions are inverted (and not investigated in the current study), and it is not mentioned internal construct validity that has to do with the factorial structure of a scale, largely investigated in the paper.

4. p.21, line 477-481: “internal consistency or single-scale instruments…. For instruments intended to have multiple factors, such…[31].” Even in the cited paper, the point is that to determine internal consistency, we need to know the internal structure of the scale. That is, if the scale has two factors, two internal consistency values should be reported (and a global value if the factors are correlated). Specifically, McDonald’s Omega is a better than Chronbach’s alpha when multiple factors have been identified.

I strongly suggest deleting these two added parts.

In conclusion, to my opinion results and discussion sections should stress that the CFA analysis did not support the intended factor structure of the scale. Thus, internal construct validity was not demonstrated. Reliability was not investigated (unless reliability indices were reported) and, however, is worthless if the scale is not valid. So I suggest to delete comments on reliability.

We look forward to receiving your revised manuscript.

Kind regards,

Francesca Chiesi

Academic Editor

PLOS ONE

---

## [Author Response · Author response to Decision Letter 2]

2 Apr 2021

Reviewer Comment 1: Whereas I can again appreciate the effort made in revising the paper, unfortunately I’m not totally convinced that the revisions help to improve the paper or to solve my previous concerns. Overall, there is still a mixed use of two distinct concepts: reliability and validity. To the best of my knowledge, reliability refers to "internal consistency" and validity refers to the "internal structure" of a scale. If the internal structure (or factor structure) is not supported by the CFA, to discuss reliability is useless. 

Response 1: We appreciate this description of the reviewer's concern. We are not clear about the reviewer's mention of "mixed use of ... reliability and validity". We have defined these terms in a manner consistent with education literature (pp.6-7 of the manuscript) and described how our data provide evidence in support or not of each concept. We have added four citations in that section as further support for our use of these definitions. While we agree that a lack of validity for an instrument would make reliability useless for the factors purported to be measured, it is possible that the instrument is reliably measuring other constructs that have not been identified in the population of interest. This information is helpful to report in the interest of future work that might attempt to identify these factors/constructs. We have also edited the Discussion (pp.28-30), Conclusion (p.31), and Future Work (p.32) sections of the manuscript to address this more explicitly.

Reviewer Comment 2: "Our characterization of CFA fit is based on published and accepted fit index cut-offs [1, 2, 3]." The fact that similar indices have been published do not mean that they are acceptable. 

Response 2: The criteria used were determined prior to analysis based on the recommendations of Jackson et al. (reported on p.28, lines 617-620) and are the cutoffs recommended in the cited textbook on measurement theory. The supporting citations of the papers in which these cutoffs are established are referenced in the Bandalos text have been cited extensively (Hu & Bentley, >75,000 citations; Browne & Cudeck, 5800 citations). We consider this to be sufficient evidence of the acceptability of these cutoffs and ask that, if the editor disagrees, he help us understand what evidence he would like to see on this front. 

Reviewer Comment 3: ML and WLS estimation methods: fit indices are not acceptable in both cases, even if slightly better for ML. However, 11 out of 20 factor loadings are <.30 if ML estimation is used. This result confirms the inadequacy of the tested factor solution. 

Response 3: Based on the cutoffs selected, RMSEA and SRMR indicate acceptable model fit under the ML estimation method, and are improved values compared to those in the analysis by Justicia and colleagues, which was deemed acceptable for the model that is now used in education research. We also respectfully direct the attention of the reviewer to Figure 3, in which the factor loadings are reported. Factor loadings for all items were greater than 0.35 using the ML method. We ultimately wish to argue that, while the fit and reliability of the model may not be good, they are at least as acceptable as other reliability results reported in the literature by those who have assessed this instrument in the past and with other populations. This underscores the importance of our validity concerns rather than undermining them. If at best the model fit is deemed acceptable and the instrument cited as reliable, the other qualitative aspects of our analysis yield sufficient concerns about validity to indicate that the survey is not measuring the constructs it is intended to measure. However, because reliability is necessary for validity, if our results do not provide sufficient reliability of the instrument due to the model fit indices, then our concerns about the validity of the results remain. We have endeavored to clarify this distinction in our paper in the CFA report (pp.21-28), Discussion (pp.28-31), and Abstract (pp.1-2).

Reviewer Comment 4: At p.6, the Section Reliability and Validity is confounding: a) "Reliability is evaluated using measures of internal consistency"': why these measures were not calculated? b) reliability has nothing to do with the "equivalence to other instruments measuring the same constructs". This aspect concerns validity. c) Criterion validity and construct validity definitions are inverted (and not investigated in the current study), and it is not mentioned internal construct validity that has to do with the factorial structure of a scale, largely investigated in the paper. 

Response 4: a) Upon further consideration, we agree that the measures of internal consistency should be reported to strengthen the analysis section and thank the reviewer for these clarifying comments. Measures of McDonald's omega have been reported in the Confirmatory Factor Analysis section of the paper (p.28, lines 623-626). b) and c) We respectfully refer the reviewer to the additions we have made that are outlined above related to new citations related to terms and their associated definitions. Equivalence is regularly used in education research to establish one part of a reliability, which does overlap with validity. Since this is a point of contention between researchers about whether this should be counted as reliability or validity, we are happy to reduce confusion among potential readers by only considering under the umbrella of validity. 

Reviewer Comment 5: p.21, line 477-481: "internal consistency or single-scale instruments…. For instruments intended to have multiple factors, such…[31]." Even in the cited paper, the point is that to determine internal consistency, we need to know the internal structure of the scale. That is, if the scale has two factors, two internal consistency values should be reported (and a global value if the factors are correlated). Specifically, McDonald’s Omega is a better than Cronbach’s alpha when multiple factors have been identified. I strongly suggest deleting these two added parts. 

Response 5: We thank the reviewer for this clarification. Coefficients omega are now reported for each factor under each CFA estimation method. 

Reviewer Comment 6: to my opinion results and discussion sections should stress that the CFA analysis did not support the intended factor structure of the scale. Thus, internal construct validity was not demonstrated. Reliability was not investigated (unless reliability indices were reported) and, however, is worthless if the scale is not valid. So I suggest to delete comments on reliability. 

Response 6: We have edited the abstract, discussion of reliability and validity, discussion, and conclusion to clarify that although the R-SPQ-2F is widely accepted as valid and reliable, perhaps it shouldn't be. We now emphasize that, although our reliability estimates are at least as good as those for other populations as reported in the literature, we nonetheless have considerable concern over validity. Our concerns over validity may reasonably extend to other populations as well since validity has never been sufficiently addressed in the literature for other populations either. We are particularly grateful to the editor for pushing on this point, as it strengthens the implications of the paper considerably.

---

## [Editor Report · Decision Letter 3]

12 Apr 2021

Validity concerns with the Revised Study Process Questionnaire (R-SPQ-2F) in undergraduate anatomy and physiology students

PONE-D-20-22094R3

Dear Dr. Johnson,

We’re pleased to inform you that your manuscript has been judged scientifically suitable for publication and will be formally accepted for publication once it meets all outstanding technical requirements.

Kind regards,

Francesca Chiesi

Academic Editor

PLOS ONE
---

## [Editor Report · Acceptance letter]

16 Apr 2021

PONE-D-20-22094R3 

Validity concerns with the Revised Study Process Questionnaire (R-SPQ-2F) in undergraduate anatomy & physiology students 

Dear Dr. Johnson:

I'm pleased to inform you that your manuscript has been deemed suitable for publication in PLOS ONE. Congratulations! Your manuscript is now with our production department. 

Kind regards, 

on behalf of

Dr. Francesca Chiesi 

Academic Editor

PLOS ONE